# Disaster Prevention Education along with Weekly Exercise Improves Self-Efficacy in Community-Dwelling Japanese People—A Randomized Control Trial

**DOI:** 10.3390/medicina57030231

**Published:** 2021-03-02

**Authors:** Akihiko Katayama, Ayako Hase, Nobuyuki Miyatake

**Affiliations:** 1Faculty of Sociology, Shikoku Gakuin University, Zentsuji-shi, Kagawa 765-8505, Japan; 2Department of Clinical Psychology, Faculty of Medicine, Kagawa University, Kita-gun, Kagawa 761-0701, Japan; ahase@med.kagawa-u.ac.jp; 3Department of Hygiene, Faculty of Medicine, Kagawa University, Kita-gun, Kagawa 761-0701, Japan; miyarin@med.kagawa-u.ac.jp

**Keywords:** disaster prevention education, self-efficacy, model course of disaster evacuation, physical fitness, physical activity

## Abstract

*Background and Objectives*: Both disaster prevention and health promotion have become public health challenges in Japan. Maintaining physical fitness from the perspective of disaster prevention and maintaining physical fitness from the perspective of health are basically covering similar issues, they are seen as different from one another because of differences in administrative jurisdiction in Japan. In the case of disaster prevention education, physical fitness is not mentioned. In and outside Japan, partial integration of disaster prevention education and health education is required. This study compares and examines the effects of disaster prevention education and traditional exercise education, as well as exercise practice. A randomized controlled trial alongside an evacuation behavior model during the event of a disaster were used in this research. *Materials and Methods*: A total of 97 community-dwelling participants were randomly allocated to two groups, a disaster prevention education group (Group D) and a traditional exercise education group (Group E). Group D received disaster prevention education with weekly exercise. Group E received traditional exercise education with weekly exercise. After ten weeks of intervention, the total evacuation time of the disaster evacuation model course, physical fitness, self-efficacy (General Self-Efficacy Scale: GSES), and health-related quality of life (QOL) were compared between the two groups. *Results*: No differences were observed between the two groups regarding the changes in the parameters of total evacuation time, physical fitness, and health-related QOL. However, the changes in GSES scores were significantly higher in Group D (1.4 ± 3.9) than in Group E (−1.1 ± 7.5). *Conclusions*: Disaster prevention education with weekly exercise significantly increased participants’ self-efficacy compared to traditional exercise education. The combination of disaster prevention education and exercise practice may have a positive effect not only on disaster prevention behavior but also on self-efficacy in health promotion. Disaster prevention education does not directly influence health promotion, but it may be a very effective method for indirectly promoting health.

## 1. Introduction

It is well known that higher physical activity and physical fitness are closely related to better health outcomes. For example, preventing lifestyle-related diseases and improving mental health. The Japanese government has advocated for increasing physical activity and physical fitness. Healthy Japan 21, which was proposed by Japan’s Ministry of Health, Labour and Welfare recommended increasing exercise habits by 36% for men aged 20 to 64 by 2023. For the same group, this campaign also recommended increasing the daily step count to 9000 steps by 2023 [1,2]. However, these goals are not easily achieved [3].

On 11 March 2011, the Tohoku Region Pacific Coast Earthquake, with a magnitude of 9.0 occurred, and the tsunami that followed destroyed many regions in the Tohoku area. A reported 15,899 people died, and 2529 people went missing, and over 90% of the deaths were due to drowning [4]. Therefore, effective disaster prevention is urgently required for all of Japan. Although many factors are involved in disaster prevention, maintaining individuals’ physical fitness, namely “disaster prevention physical fitness” is thought to be important [5].

To prevent lifestyle-related diseases and maintain mental health, increasing physical activity and physical strength are necessary [3]. Also, to protect lives in a disaster, these components are essential in maintaining and improving physical strength [5]. These issues are related in that they involve human physical activity. However, in Japan, these are considered entirely different matters. Japan’s Ministry of Health, Labour, and Welfare is responsible for public health measures such as preventing lifestyle-related diseases. The Cabinet Office and the Ministry of Land, Infrastructure, Transport, and Tourism are in charge of disaster prevention measures such as disaster countermeasures. 

Reports have shown that there are problems in cooperation between professionals in the field of disaster prevention not only in Japan but also internationally [6]. These reports indicate that each professionals’ members have created a new educational framework for disaster medicine and public health preparedness for collaboration between professionals [7]. The necessity of an educational program for facilitating cooperation between each ministry and agency has been reported, and the educational program has been proposed [8]. In this way, in the field of disaster prevention, cooperation with other fields is being promoted. However, the effect of cooperation has not yet been clarified.

Based on these issues, we would like to offer certain proposals for more efficient health promotion by daringly combining disaster prevention and health promotion perspectives. We hypothesized that disaster prevention education, in addition to weekly exercise, may be beneficial for improving physical fitness and mental health, including self-efficacy. Therefore, in this study, we investigated the effect of disaster prevention education in addition to weekly exercise on physical fitness and mental health in community-dwelling Japanese people using a randomized controlled trial (RCT). Our objective was to evaluate the efficacy of disaster prevention education for health promotion.

## 2. Materials and Methods

### 2.1. Participants

The recruitment of participants was carried out with the cooperation of the government (Marugame City, Japan), and their public relations magazine published the recruitment descriptions. We asked the participants to mail us a request form. Then, we held a research participation briefing session and thoroughly explained the research contents. One hundred four people wanted to participate in the briefing session. After the briefing session, seven people refused to participate. Six people were abscent because of opposition from their family to join (personal circumstances). And one person did not participate due to health reasons.

A total of 97 community-dwelling Japanese individuals among 104 participants, aged 70.6 ± 5.1 years and who met the following criteria were enrolled in the RCT study. The participants all met the following criteria: (1) they voluntarily took part in the study in May 2018 in Marugame city, Kagawa prefecture, Japan, (2) they had no limitations on exercise placed on them by medical doctors; and (3) they submitted written informed consent.

Ethical approval was obtained from the ethical committee of Shikoku Gakuin University, Zentsuji city, Kagawa prefecture, Japan (approval number: 2018001, approval date: 1 August 2018).

### 2.2. Study Design

The design of this study used a Randomized Controlled Trial (RCT) (Figure 1). The 97 enrolled participants were allocated to two groups: a disaster prevention education group (Group D; n = 49) and an exercise education group (Group E; n = 48), with sex-stratified randomization. Group D received disaster prevention education for 30 min a week for four weeks. The disaster prevention education consisted of the following: necessary information on disaster occurrence mechanisms, disaster occurrence situation in Japan, disaster preparation, and evacuation behavior. Group E received general exercise education for 30 min a week for four weeks. General exercise education consisted of information on the prevention of metabolic syndrome, locomotive syndrome, and dementia. All participants in both groups followed the same physical fitness curricula, which consisted mainly of aerobic exercise and strength training by well-trained physical fitness instructors, for 90 min a week for ten weeks.

We performed measurements before and after the intervention. All baseline measurements were completed one week before the intervention. All measurements were completed one week after the end of the intervention. 

The disaster evacuation model course’s total time was set as the primary evaluation item, and the physical fitness was set as the secondary evaluation item. We also measured self-efficacy (General Self-Efficacy Scale: GSES) and health-related quality of life (QOL).

### 2.3. Sample Size

The primary endpoint was the total evacuation time on a model evacuation course in a disaster. We compared the change in measured values before and after the intervention between the two groups. We calculated the sample size using the significance level, power, difference between the two groups, and standard deviation in an unpaired *t*-test. 

The difference between the two groups was derived as follows. The total evacuation time was estimated based on a pilot study involving 72 participants. From the previous study, the changes and standard deviation of the 6-min walk due to the intervention were used as a reference to estimate the changes and standard deviation of the total evacuation time due to the intervention. The difference in change in whole evacuation time between the two groups was set to 15 s in this study. Similarly, the standard deviation was calculated to be 25. Following similar previous studies, the significance level used was α: 0.05; the power used was 0.8. 

Sample size was tested by unpaired *t*-test (significance level: α = 0.05, power = 0.8, detected difference: 15, standard deviation: 25, case ratio. 1). The sample size was calculated using JMP^®^ 12 software (SAS Institute Japan, Tokyo, Japan). Then, the estimated sample size was 90 participants (two groups, each with 45 participants). The dropout rate was estimated at 10%. Finally, it was calculated that 100 participants (two groups, each with 50 participants) were needed for the study.

### 2.4. Clinical Parameters and Measurements

The parameters of sex, age, height (cm), and body weight (kg) were obtained for every patient. Each patient’s body mass index (BMI) was calculated as follows: body weight (kg)/(height (m))^2^.

### 2.5. Self-Efficacy

The concept of self-efficacy was first proposed by Bandura [9]. In his social learning theory, three factors, namely a precedent factor, a result factor, and a cognitive factor, are considered to decide the person’s action. Self-efficacy becomes the main element of the precedent factor of the movement of the person [9]. GSES (general self-efficacy scale) measures personal general self-efficacy with high reliability and validity, as previously described [10,11]. Participants with a higher GSES score can adapt their behavior, work diligently, and be more tolerant [12].

The GSES can quantify self-efficacy and compare its levels [13]. The GSES consists of 16 questions in a “yes” or “no” two-point system, with a score range of 0–16 points. The measured scores are returned as standardized scores using the GSES standardized score conversion table. The standardized score is the same as the deviation score, with a mean of 50 and a standard deviation of 10. The higher this standardized score is, the higher the self-efficacy is [13].

### 2.6. Health-Related Quality of Life (QOL)

SF-36^®^ was used to measure the health-related QOL changes in the participants after the intervention. SF-36^®^ is a self-report-type health condition questionnaire. It measures a comprehensive health concept using eight domains. SF-36 consists of 36 items that can be answered in around five minutes. SF-36 can measure health degree [14,15]. The SF-36v2^®^ Japanese edition, which improved the original SF-36, was used as Standard Edition in this study [16,17]. There are three components: “the role/social side”, “the physical side”, and “the mental side”. In this study, we use these three summary scores as an outcome [18].

### 2.7. Model Course of Evacuation at Disaster Onset

The model course of evacuation at disaster onset with a tsunami was established at physical education facilities at Marugame City (Kagawa Prefecture, Japan). An evacuation model course was set with reference to the tsunami evacuation tower in Nankoku City (Kochi Prefecture, Japan). We collected information on evacuation routes from local disaster management officials and local disaster management leaders. We also measured some actual evacuation courses for residents at the site. The measured items were the moving distance, moving time, expected obstacles, and height of climbing.

The model course consisted of three stages directed by experts in disaster prevention. The first stage was a horizontal evacuation of 100 m, the second stage was an obstacle evacuation of 80 m, and the third stage was a vertical evacuation of 12 m. In the first stage, the participants were directed to walk on flat ground. In the second stage, they were told to walk in such a way as to avoid the obstacles. In the third stage, we were instructed to go up and down the stairs as high as the 12m high tsunami evacuation facility in Nankoku City. During the measurements, the participants were directed to carry 5 kg bags on their back under close monitoring by the observer to prevent them from falling. We measured the time of horizontal evacuation, obstacle evacuation, vertical evacuation, and total evacuation (seconds).

### 2.8. Physical Fitness

Physical fitness was evaluated by grip strength (kg), flexibility (cm), one leg with open eye balance (seconds), 6-min walking (meters), and 10 meter walking with obstacles (seconds), according to the new physical fitness test by Japan’s Ministry of Education, Culture, Sports, Science, and Technology [19,20]. In addition, the participants took the 30-s chair standing up test [21,22], five times chair standing up test [23,24], and locomotive syndrome test established by The Japanese Orthopaedic Association [25].

### 2.9. Statistical Analysis

The data were expressed as mean ± standard deviation in continuous valuables and number (proportion: %) in categorized valuables. An unpaired t-test was used for comparison between the two groups, where *p* < 0.05 was significant. Statistical analysis was performed using JMP^®^14 software (SAS Institute Japan).

## 3. Results

### 3.1. Clinical Characteristics of Enrolled Participants at Baseline

For the four weeks of education and 10 weeks of exercise, three participants in Group D and four participants in Group E had a low participation rate (<80%). One participant in each group dropped out. Therefore, for the final analysis, we evaluated 45 participants in Group D and 43 participants in Group E after 10 weeks of exercise. The clinical profiles of the enrolled participants (45 participants in Group D, 43 participants in Group E, and 88 participants in total) at baseline are summarized in Table 1.

### 3.2. Comparison of the Changes in Parameters between the Two Groups

We compared changes in the parameters between the two groups (Table 2). No difference was observed in the changes in the three-stage total evacuation times between the groups after 10 weeks of intervention. No differences were observed between the two groups regarding changes in the horizontal evacuation time, obstacle evacuation time, or vertical evacuation time. Also, there were no significant differences between the two groups in terms of the physical fitness test index. Similarly, the three values of SF36, a measure of health-related QOL, did not differ significantly.

However, the changes in self-efficacy evaluated by GSES scores in Group D (1.4 ± 3.9) were significantly higher than those in Group E (−1.1 ± 7.5) (*p* = 0.036). No significant differences in the changes in other clinical parameters were observed between the two groups.

## 4. Discussion

In this study, we evaluated the effects of disaster prevention education in addition to weekly exercise on physical fitness by RCT. There were no significant changes in physical fitness due to the intervention, but self-efficacy improved significantly.

Many practical research studies have been conducted and reported on disaster prevention education. Torani et al. summarized 11 studies on disaster education and reported that disaster education is a functional, operational, and cost-effective tool for risk management [26]. Aghaei et al. reported the major categories of disaster education strategies such as raising knowledge, educational planning, educational approaches, from the perspective of the importance of educational strategies for disaster risk reduction [27]. Thus, there are many research reports on the importance of disaster education and its effectiveness. Based on this evidence, practical disaster education is being conducted in local communities and other organizations. However, no studies have researched the relationship between disaster prevention education, physical fitness, and health promotion. To the best of our knowledge, this is the first study on the relationship between disaster prevention education, physical fitness, and health promotion.

There are many reports on how to improve physical activity and physical fitness. However, the persistence rates were not comparably high, and the long-term effects were not established. Therefore, various strategies for improving physical activity and physical fitness in clinical practices were considered. Foley et al. reported that changing commuting methods such as cycling was associated with increases in the amount of physical activity, indicating that attention to commuting means may be one of the fundamental approaches for increasing physical activity [28]. Smith et al., in their analysis, investigated the association between environment and physical activity, and they found that providing appropriate parks and playgrounds, as well as improving public transport, had a positive impact and increased walking [29]. Althoff et al. reported that game apps with physical activity, such as Pokémon Go, increased short-term physical activity [30]. Furthermore, using health and fitness applications improved physical activity and sedentary behavior in 27 studies, including 19 RCTs [31]. In this study, we used disaster prevention education in addition to a weekly exercise. There was no significant difference in the physical fitness parameters between the two groups, but self-efficacy showed a significant difference.

Concerning the relationship between self-efficacy and physical activity and/or physical fitness, Shieh et al.’s cross-sectional study reported that self-efficacy was significantly associated with exercise habits [32]. Lewis et al. also demonstrated a relationship between self-efficacy and exercise habits by RCT [33]. Moreover, in a meta-analysis, effective interventions increased both self-efficacy and physical activity [34]. Taken together, these findings indicate that self-efficacy is closely associated with increases in physical activity and/or fitness. People with high self-efficacy can set clear goals. Therefore, to increase physical activity and/or physical fitness, it may be necessary to improve a person’s self-efficacy first.

In this study, we mainly evaluated the effect of disaster prevention education with weekly exercise on total evacuation time using a model course of evacuation at disaster onset. There was no significant difference in the change in total evacuation time or physical fitness between Group D and Group E. However, self-efficacy was significantly improved. Other strategies, including disaster prevention education rather than education such as healthy exercise, may be beneficial in improving health-related self-efficacy. From a long term perspective, improving self-efficacy may improve physical activity and/or physical fitness. Therefore, disaster prevention education with weekly exercise may be beneficial for physical activity and/or physical fitness in clinical practice.

In Japan, health-related fields and disaster prevention are considered entirely different matters, and separate departments are responsible for their administrative jurisdiction [35]. However, this study suggests that both parties may interact better to achieve better results.

Incorporating disaster prevention education as a part of health measures may enhance the effectiveness of health policies for residents. Accurate preparation for disasters that are expected to occur in the future may improve the self-efficacy of residents and naturally lead them to practice physical fitness. It may be possible to lead people to increase their physical activity and practice exercise through the indirect method of disaster prevention, rather than the method of emphasizing the need for health and forcing behavior change. It is also important to consider disaster prevention as one way to improve health, rather than simply as an activity to protect oneself from disasters.

There are some limitations to our research. First, since the intervention period was only 10 weeks, it was impossible to evaluate the long-term effects of disaster education on physical fitness, quality of life, and self-efficacy. Second, the study’s statistical power may be slightly lower than the original estimate because only 45 participants in group D and 43 participants in group E were evaluated. Third, enrolled Participants in this study were thought to be more health-conscious than the average person.

## 5. Conclusions

Disaster prevention education with weekly exercise significantly improves self-efficacy compared with conventional exercise education with weekly exercise. Improving self-efficacy may increase appropriate behavior, like physical activity and/or physical fitness. Therefore, the practice of disaster education may finally lead to increased physical activity and/or physical fitness. Disaster prevention education does not directly influence health promotion, but it may be a very effective method for indirectly promoting health.

## Figures and Tables

**Figure 1 medicina-57-00231-f001:**
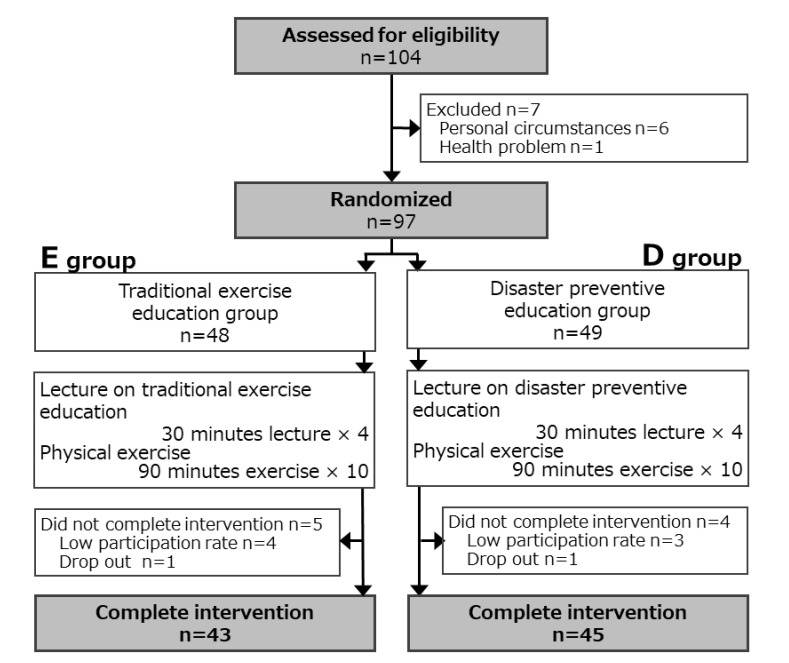
Flow chart of the study design.

**Table 1 medicina-57-00231-t001:** Baseline characteristics of subjects.

	D: Disaster Preventive Education Group (n = 43)					E: Traditional Exercise Education Group (n = 45)				
	mean	±	SD	min	max	mean	±	SD	min	max
Male, n (%)	7 (16.2%)					7 (15.6%)				
Age (year)	70.3	±	5.3	56	86	71.0	±	4.8	63	83
Height (cm)	156.5	±	7.1	144.0	174.0	153.6	±	5.8	139.0	166.0
Body weight (kg)	55.2	±	7.3	38.0	76.0	52.3	±	7.8	40.0	72.0
BMI (kg/m^2^)	22.5	±	2.5	15.2	29.4	22.1	±	2.7	16.4	29.6
Grip strength (kg)	26.1	±	6.5	14.3	47.4	23.8	±	5.8	14.7	40.4
Flexibility (cm)	35.1	±	9.0	15.0	52.5	34.2	±	8.7	18.0	54.0
One leg with eye opened balance (seconds)	86.6	±	39.6	12	120	92.0	±	36.9	9	120
Six minutes’ walking (m)	585.2	±	70.9	280	760	566.1	±	65.2	310	760
10 meters’ walking with obstacles (seconds)	6.3	±	1.0	4.0	7.9	6.7	±	1.3	4.8	12.5
30 seconds chair standing up test (times)	21.4	±	5.4	11	36	22.6	±	5.5	12	40
Five times chair standing up test (seconds)	5.9	±	1.3	3.8	10.6	6.0	±	1.3	4.3	9.4
Two-step test	1.41	±	0.14	1.07	1.65	1.39	±	0.18	1.00	1.70
Horizontal evacuation (seconds)	71.9	±	7.1	55	90	72.7	±	8.7	58	90
Obstacle evacuation (seconds)	71.9	±	10.7	52	101	70.5	±	14.4	41	113
Vertical evacuation (seconds)	89.3	±	12.6	72	130	90.2	±	20.2	48	166
Total evacuation (seconds)	233.1	±	24.6	190	306	233.4	±	38.1	160	369
SF-36 PCS	47.2	±	8.3	32.1	63.0	46.3	±	10.3	9.2	61.6
SF-36 MCS	53.2	±	7.5	36.2	69.0	54.4	±	6.9	39.3	66.8
SF-38 RCS	49.6	±	10.7	20.0	64.2	48.8	±	10.3	15.1	66.2
GSES score	50.81	±	10.35	29	68	50.18	±	9.72	32	66

BMI: Body mass index (kg/m^2^); Two-step test: Locomotive syndrome risk test; SF-36 PCS: SF-36 Physical Component Summary; SF-36 MCS: SF-36 Mental Component Summary; SF-36 RCS: SF-36 Role-social Component Summary; GSES: General Self-Efficasy Scale.

**Table 2 medicina-57-00231-t002:** Comparison of changes between groups.

	D: Disaster Preventive Education Group (n = 43)			E: Traditional Exercise Education Group (n = 45)			
	mean	±	SD	mean	±	SD	*p-*Value
⊿Grip strength (kg)	−0.5	±	2.8	0.3	±	1.8	0.07
⊿Flexibility (cm)	0.7	±	5.3	1.5	±	5.5	0.44
⊿One leg with eye opened balance (seconds)	2.9	±	28.9	4.0	±	23.4	0.90
⊿Six minutes’ walking (m)	29.4	±	49.8	27.1	±	41.3	0.71
⊿10 meters’ walking with obstacles (seconds)	−0.6	±	0.7	−0.8	±	0.6	0.33
⊿30 seconds chair standing up test (times)	6.99	±	5.55	6.51	±	4.33	0.51
⊿Five times chair standing up test (seconds)	−0.50	±	1.30	−0.93	±	0.77	0.13
⊿Two-step test	0.12	±	0.15	0.17	±	0.13	0.08
⊿Horizontal evacuation (seconds)	−15.36	±	10.91	−13.31	±	9.76	0.13
⊿Obstacle evacuation (seconds)	−9.65	±	12.04	−12.69	±	14.66	0.51
⊿Vertical evacuation (seconds)	−4.57	±	13.88	−5.09	±	9.76	0.57
⊿Total evacuation (seconds)	−30.27	±	20.52	−31.09	±	20.33	0.54
⊿SF-36 PCS	0.32	±	8.57	0.74	±	6.30	0.97
⊿SF-36 MCS	2.74	±	7.36	0.16	±	6.39	0.11
⊿SF-38 RCS	−0.29	±	7.97	−1.28	±	10.66	0.84
⊿GSES score	−1.11	±	7.50	1.40	±	3.94	**0.04**

Two-step test: Locomotive syndrome risk test; SF-36 PCS: SF-36 Physical Component Summary; SF-36 MCS: SF-36 Mental Component Summary; SF-36 RCS: SF-36 Role-social Component Summary; GSES: General Self-Efficacy Scale; Bold values are statistically significant (*p* < 0.05).

## Data Availability

Not applicable.

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
