# Peer review of "Disaster Prevention Education along with Weekly Exercise Improves Self-Efficacy in Community-Dwelling Japanese People—A Randomized Control Trial"

_medicina, 2021, doi:10.3390/medicina57030231_

Round 1

Reviewer 1 Report

Overall, the study was adequate, well-written, and presented. It might be strengthened by discussing the theoretical approach more clearly in the introduction to make a case for the study. Results were modest raising the question of the significance of the study, but they were well presented.

Author Response

Thank you very much for reviewing our paper.

We have fully considered your suggestions and have revised our paper.

I would appreciate it if you could recheck it.

Overall, the study was adequate, well-written, and presented. It might be strengthened by discussing the theoretical approach more clearly in the introduction to make a case for the study. Results were modest raising the question of the significance of the study, but they were well presented.

In the introduction section, we have added examples from Japan as well as overseas.

We have tried to strongly explain the purpose and importance of this study based on these examples.

We have not changed the result section, but we have added a sentence in the conclusion section that includes the content we wanted to convey.

We hope you will check it out.

Reviewer 2 Report

Totally the present article is well-established, and the subject is interesting, but some major revision should be considered. I like the method section which is very solid

ABSTRACT

The abstract should state briefly the purpose of the research, the principal results and major conclusions. An abstract is often presented separately from the article, so it must be able to stand alone.

INTRODUCTION

In the introduction there is still lacks the international characteristics. A strong background of the known knowledge is needed (The necessity and innovation of the article should be presented to the introduction. It is suggested to present the structure of the article at the end of the introduction.)

DISCUSSION

It is suggested to compare the results of the present research with some similar studies which is done before.

CONLUSIONS

In addition, a closing paragraph or two with summative insights, or overarching principles garnered from this paper would make the manuscript more complete

 REFERENCES

Following, you will find some important relatead references which should be added to the list:

  • Khorram-Manesh A, et al. Education in disaster management: what do we offer and what do we need? Proposing a new global program. Disaster medicine and public health preparedness. 2016 Dec 1;10(6):854-73.
  • Goniewicz K, et al. The importance of pre-training gap analyses and the identification of competencies and skill requirements of medical personnel for mass casualty incidents and disaster training. BMC public health. 2021 Dec;21(1):1-1.
  • Subbarao I, et al. A consensus-based educational framework and competency set for the discipline of disaster medicine and public health preparedness. Disaster medicine and public health preparedness. 2008 Mar;2(1):57-68.

Some additional copy editing, both to smooth out the areas of clumsy prose due to translation and to use translated terms that are consistent with current medical vernacular will also be required

Author Response

Thank you very much for reviewing my paper.

I have restructured my paper according to your guidance.

I understand the importance of the abstract again.

I would appreciate it if you could recheck it.

ABSTRACT

The abstract should state briefly the purpose of the research, the principal results and major conclusions. An abstract is often presented separately from the article, so it must be able to stand alone.

Thank you very much for your precise guidance on this essential matter.

We have restructured the abstract.

We gave more detail in the background and conclusions so that the study could be understood from the abstract alone.

INTRODUCTION

In the introduction there is still lacks the international characteristics. A strong background of the known knowledge is needed (The necessity and innovation of the article should be presented to the introduction. It is suggested to present the structure of the article at the end of the introduction.)

Thank you for pointing this out.

We have rechecked the introduction part and found that we lacked an international perspective.

We have used the references you provided to introduce cases outside of Japan and explain the significance of this study.

Please check it out.

DISCUSSION

It is suggested to compare the results of the present research with some similar studies which is done before.

Our paper was missing some important details.

Thank you for pointing it out.

We searched again for similar studies with the same purpose as this study but could not find any papers on the theme of disaster education and physical fitness.

Therefore, we used a paper reporting on the current status of disaster prevention education to explain that the perspective of physical fitness has not yet been established in disaster prevention education. I added this content to the discussion section.

Using these explanations, we explained the originality of our study.

CONCLUSIONS

In addition, a closing paragraph or two with summative insights, or overarching principles garnered from this paper would make the manuscript more complete.

Thank you very much for guiding us with your very essential thoughts on the Conclusion section.

With your guidance, we were able to add what we really wanted to say in this paper.

Please check the last sentence that we added.

REFERENCES

Following, you will find some important related references which should be added to the list:

Some additional copy editing, both to smooth out the areas of clumsy prose due to translation and to use translated terms that are consistent with current medical vernacular will also be required.

Thank you very much for providing us with very effective references.

We have used these references in the introduction section.

For the paper we are going to resubmit, we went through English editing and grammar checking again.

Round 2

Reviewer 2 Report

I find the paper improved. Thanks for taking my suggestions into consideration.